# Impact Assessment of the Implementation Effect of the Post-Relocation Support Policies of Rural Reservoir Resettlers' Livelihoods in Energy Transition

Bing Liang [1,2,3,*], Guoqing Shi [1,2,3,*] , Zhonggen Sun [1,2,3], Yuelin Wang [1,2,3], Bosen Zhang [4,*], Yuangang Xu [1,5] and Yingping Dong [6,7]

1 National Research Center for Resettlement, Hohai University, Nanjing 211100, China; sunzhonggen@sina.com (Z.S.)
2 Institute of Social Development, School of Public Administration, Hohai University, Nanjing 211100, China
3 Asian Research Center, Hohai University, Nanjing 211100, China
4 College of Hydrology and Water Resources, Hohai University, Nanjing 211100, China
5 Provincial Ecological Resettlement Bureau of Guizhou, Guiyang 550001, China
6 School of Geography & Environmental Science, Guizhou Normal University, Guiyang 550001, China
7 Engineering Technology Institute for Karst Desertification Control, Guizhou Normal University, Guiyang 550001, China
* Correspondence: bliang1@126.com (B.L.); gshi@hhu.edu.cn (G.S.); 21021412001@hhu.edu.cn (B.Z.)

**Abstract:** Energy transition is a major structural change in the whole social system, and the energy system must be changed globally to replace fossil fuels. Hydropower is one of the largest sources of renewable energy in the world. However, owing to the construction of hydropower projects, involuntary resettlers are suffering from being far away from their native land, losing the land cultivated for generations and the houses they live in, and losing the social relationship network based on geography and blood ties. Based on the system evaluation theory of reservoir resettlement and referring to relevant research findings, this paper constructs a comprehensive evaluation index framework for assessing the implementation effect of the Post-Relocation Support (PReS) policy. The research region is located in Zhijin County, Bijie City, Guizhou Province, China. Accordingly, a combined method of a structural equation model and a fuzzy comprehensive evaluation model is used in this paper to analyze the implementation of the PReS policy. The results show that the total score of implementing effects of the PReS policy is 4.4, with dramatic significance. The subindex scores of the resettlers' family income, living conditions, and production conditions; the local economy; and social stability are 4.3, 4.6, 4.4, 4.6, and 4.3, respectively, with dramatic significance. This paper has analyzed and summarized the successful practices of implementing the PReS policy for reservoir resettlers in three dimensions: poverty alleviation, beautiful home construction, and accelerating rural revitalization. Research shows that China's rural reservoir resettlers' PReS policy has been more effective in restoring the livelihoods of reservoir resettlers.

**Keywords:** rural reservoir resettlement; Post-Relocation Support (PReS) policy; resettlers' livelihoods; impact assessment; structural equation model; fuzzy comprehensive evaluation method; energy transition

## 1. Introduction

Energy transformation is a major structural change in the energy system. Historically, these changes have been driven by the demand for different fuels and the availability of fuels. Utilization of hydropower resources is an important method of renewable and clean energy transformation and utilization that is also able to effectively reduce carbon emissions [1]. To use hydropower resources to reduce carbon emissions, it is often necessary to manually build hydraulic structures, such as dams and diversion pipe culverts, allowing the concentration of water drops and regulating flow [2]. On this basis, the following

advantages of hydropower generation are recognized: high-power generation efficiency, low cost, fast unit startup, and easy regulation [3]. China has completed the majority of its hydropower projects, and solar, wind and even hydrogen will be the sources of China's energy in the next generation, which is more important for China's just transition. As an important part of the new power system, pumped storage plays a significant role in the development of clean and low-carbon energy transition and represents an important initiative in terms of building a new power system, promoting sustainable development, and achieving the goal of peak carbon neutrality. Constructing hydropower projects leads to involuntary resettlers and even sacrifices the interests of some involuntary resettlers by forcing them to leave their hometowns; some resettlers lose land and houses, some lose social networks, and some fail to adapt to the production and life of resettlement sites. Involuntary resettlers are a worldwide problem, and compensation alone cannot fulfill the goals of livelihood restoration, income improvement, life improvement, and shared development benefits for resettlers [4–6]. Reservoir resettlers are involuntary resettlers, and supporting their livelihood is bound to be a long-term, multistage, and arduous process [7,8].

In the relevant report of the World Commission on Dams (WCD), those who are negatively affected by the development and utilization of water resources (such as the construction of hydropower stations) are called displaced people, resettlers, or affected people. The activities of these people are divided into displacement, resettlement, rehabilitation, repair, and development; that is, DRRD [9–11]. DRRD represents the whole resettlement process, which includes various risks and may even lead to poverty [12]. The original life system of reservoir resettlers will be destroyed; economic activities and income will be interrupted [13]; and clothing, food, housing, and transportation will not be guaranteed. Additionally, reservoir resettlers may lose necessary social public services, such as medical care and education; deviate from the mainstream of society; be marginalized; and even become refugees [14,15]. Michael Cernea, a famous sociologist at the World Bank, proposed the Impoverishment Risks and Reconstruction (IRR) model to describe the inherent risks of involuntary resettlement. According to the IRR model, involuntary resettlement includes eight key poverty risks: land loss, unemployment, loss of home, marginalization, unsafe food, disease, loss of access to public property and services, and community disintegration [16], an Indian scholar, fully verified the feasibility of the IRR model by studying and analyzing the data of Indian resettlers over the past 20 to 30 years. He evaluated the Post-Relocation Support (PReS) policy for Indian resettlers with the help of the IRR model and pointed out that the focus of the PReS policy for Indian resettlers is to prevent poverty caused by land loss [17]. There are many reasons for the above problems in resettlement practice; some of these problems are related to the resettlers themselves, some to the planning and management work, and some to resettlement policies and regulations [18]. To solve the abovementioned resettlement problems, the World Bank issued a guiding document on resettlement in 1990. The document, titled the World Bank working guidelines: involuntary resettlement (operational guidelines OD4.30), pointed out that priority should be given to the resettlement issue and that the best scheme should be selected in the early planning stage of the project [19]. The inadequacy of the World Bank document is that it advises that the compensation claim for the resettlers be paid in full before the relocation and that compensation not involve assistance for the problems and difficulties faced by the resettlers after the relocation [20]. The resettlement policy focuses mainly on project construction while ignoring concerns about the development of resettlers, resulting in insufficient resettlement funds and prominent environmental problems [21]. Cernea, a research expert at the World Bank, pointed out that the purpose of resettlement policy formulation should be to help resettlers recover and improve their living conditions [22]. Therefore, to further properly address the livelihood restoration of involuntary resettlers, the World Bank subsequently issued the World Bank's work guide on involuntary resettlement in 1995. This document specifies the policy objectives, planning points, tasks, and project selection requirements of involuntary resettlement in detail [23]. The main highlight is that the policy formulation fully considers the people-oriented princi-

ple (such as minimizing the number of resettlers and transforming temporary compensatory and relief measures for resettlers into comprehensive development-oriented solutions), and all projects involving resettlement are covered [24,25].

In fact, the impact of relocation on the social system of resettlers cannot be completely avoided [26]. As long as there is project construction, a World Bank study on reservoir resettlement focuses mainly on examining the early compensation and relocation of re-settlers and advocates giving one-time compensation to resettlers before relocation [27]. However, the study pays insufficient attention to PReS for reservoir resettlers and rarely involves application analysis, especially research on the economy, society, culture, and other aspects closely related to reservoir resettlers [28]. China has carried out large-scale water conservancy and hydropower projects for social and economic development. Many reservoir, embankment, river, and canal construction projects have led to widespread resettlements. The population of reservoir resettlers alone has reached more than 20 million. China is the country with the most involuntary resettlers in the world, so China is very representative of the research on the just transition of migrant livelihoods [29,30]. To help the reservoir resettlers eliminate poverty and prosper and to help promote the economic and social development of the reservoir area and the resettlement area, ensure the healthy development of water conservancy and hydropower work in China in the new period, and build a socialist harmonious society, China issued the opinions of the State Council on improving PReS for large and medium-sized reservoir resettlers in 2006 and established the late stage support fund for large and medium-sized reservoir resettlers and the reservoir area fund for large and medium-sized reservoirs. For the resettlers of large and medium-sized reservoirs, the unified support is 20 years, and each person's support 600 yuan per year. The use of the support funds is determined according to the principle of "one try and two can": "one try" means that the government should try to provide resettlers directly with cash compensation as much as possible; "two can" means a combination of cash compensation and project support. This study attempts to solve the problems of reservoir resettlers, especially the livelihood restoration of reservoir resettlers in poor areas, by constructing an impact assessment system of the PReS policy effects on rural reservoir resettlers; this system uses a structural equation model to determine the weight of impact assessment indicators and uses a mathematical fuzzy comprehensive evaluation method to evaluate the PReS policy for Chinese resettlers.

## 2. Research Method

### 2.1. Construction of the Impact Assessment Index System

In May 2011, the National Development and Reform Commission, the Ministry of Finance, and the Ministry of Water Resources jointly issued a notice on monitoring and evaluating the implementation of the PReS policy for large and medium-sized reservoir resettlements (this notice is hereafter referred to as Document No. 1033) after expert consultation and interministerial joint meetings widely solicited opinions throughout the country. Document No. 1033 clearly stipulates the impact assessment indicators of the implementation effects of the PReS policy; these indicators include 5 secondary indicators and 14 tertiary indicators. See Figure 1 for details.

This study has fully considered the requirements of impact assessment objectives in building a comprehensive impact assessment index system for the implementation effects of the PReS policy. Under the framework of Document No. 1033 on impact assessment indicators for the implementation effects of the PReS policy and the impact assessment theory of the rural reservoir resettlement system, combined with the actual situation of the surveyed area and the research results of experts and scholars in this field, it also follows the general principles of index selection and the specific principles related to the research topic. The implementation effects of the PReS policy should be evaluated from multiple angles [31]. Figure 2 shows the comprehensive impact assessment index system of the implementation effects of the PReS policy constructed in this study, with 5 secondary indicators and 17 tertiary indicators established.

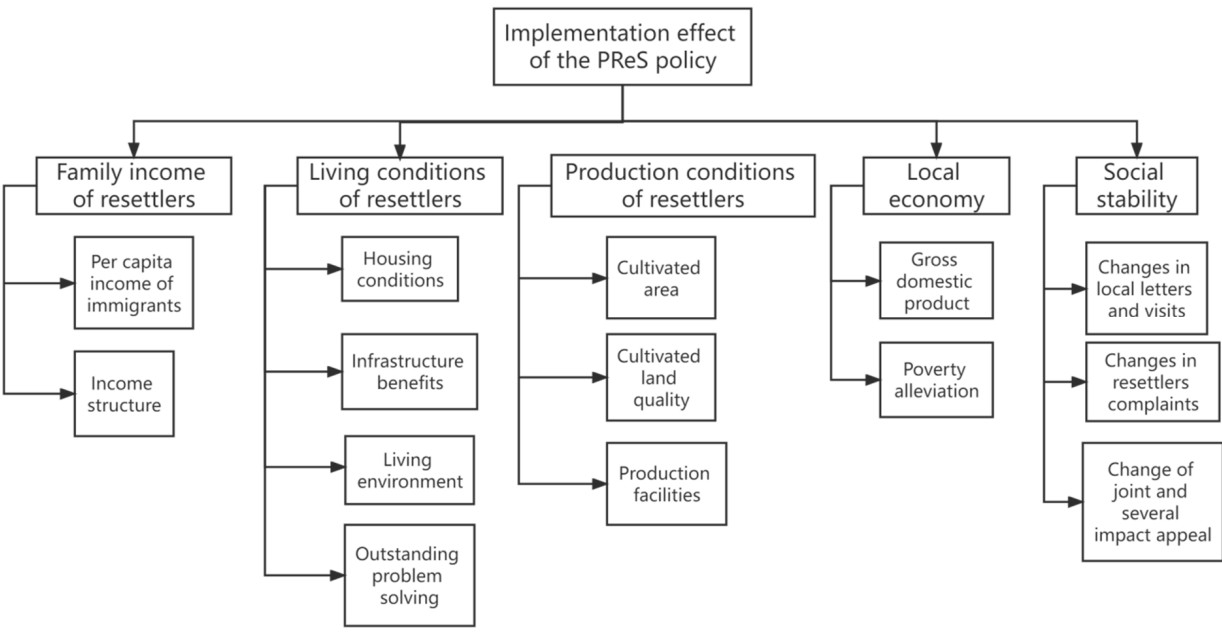

**Figure 1.** Indicator system for the impact assessment of the PReS policy.

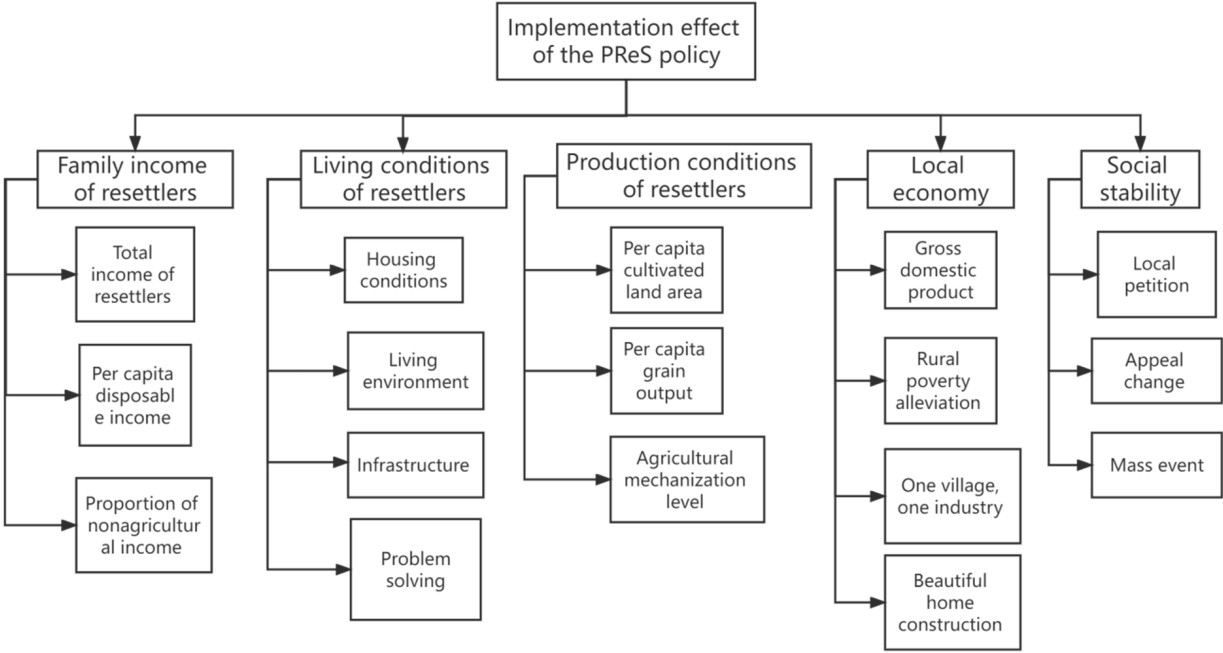

**Figure 2.** Impact assessment index system of the implementation effects of the PReS policy.

*2.2. Research Areas*

In this paper, Zhijin County is selected as the survey sample to study the impact assessment of the PReS policy. Zhijin County is located in west-central Guizhou Province, south of the Bijie Experimental Area, and in the triangle at the intersection of the Liuchong and Sancha Rivers, the tributaries of the Wujiang River, as shown in Figure 3. With a total area of 2865.28 square kilometers, the county administers 578 villages (communities) in 33 townships (subdistricts) and is inhabited by 1.256 million people from 26 ethnic groups, including Han, Miao, Yi, Bai, and Buyi. Zhijin County is an important part of the "Qianzhong economic zone". Zhijin County has 6 large and medium-sized reservoirs, including 5 large reservoirs and 1 medium-sized reservoir. Zhijin County began to investi-

gate and register the immigrant population near large and medium-sized reservoirs in the county in July 2006, with a total of 17,154 people registered.

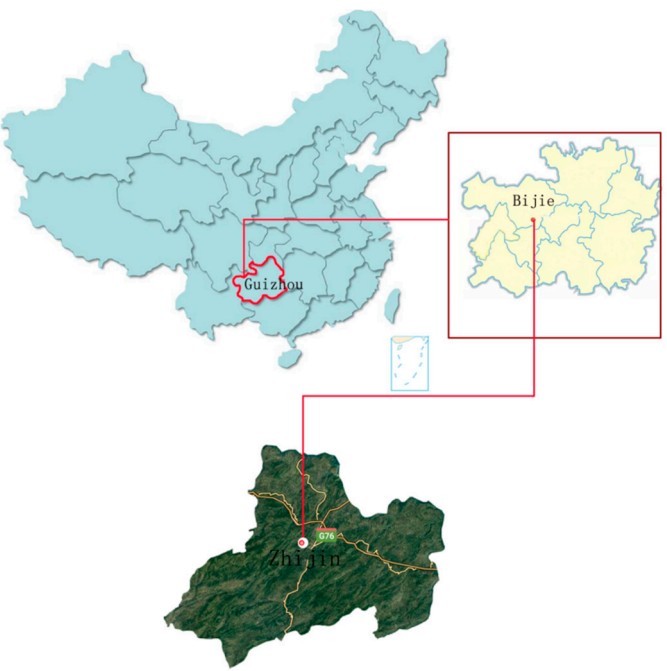

**Figure 3.** District map of Zhijin County, Bijie City, Guizhou Province, China.

Zhijin County is selected as the survey sample because, on the one hand, Zhijin County has many PReS resettlers. In 2021, the county comprised 24 towns and 483 villages and had 785 groups, 5077 households, and 25,681 people. Zhijin County used to be deeply poverty-stricken. Many resettlers are both the supporting population and the poor population. Their multiple identities are intertwined. In the post-poverty-alleviation era, how to realize the just transition of resettlers' livelihood is a good breakthrough from the perspective of the implementation effects of the supporting policy for rural resettlers against the background of China's rural revitalization. It is typical and representative to select Zhijin County as the survey sample.

*2.3. Data Source*

Questionnaire Design

To use a structural equation model to analyze the implementation effects of the PReS policy, this study, based on the comprehensive impact assessment index system of the implementation effects of the PReS policy, selects the 5-point scale method and designs 17 indicator options, which mainly reflect the impact assessment of the survey objects on the implementation effects of the PReS policy. The scale includes these five options: 1 is "significantly reduced", 2 is "slightly reduced", 3 is "no change", 4 is "slightly increased or improved", and 5 is "significantly increased or improved". The five options are defined in Table 1.

From 21 December 2021 to 20 January 2022, with the support and cooperation of the People's Government of Zhijin County and the Rural Revitalization Bureau, we carried out field surveys, questionnaire surveys, household interviews, township interviews, and county-level data collection in sample villages. We strove to make the questionnaire for evaluating the PReS policy objective and fair. We distributed questionnaires to reservoir resettlers, ordinary farmers, village cadres, reservoir resettlement cadres at all levels, county and township cadres, and other stakeholders. A total of 372 questionnaires were distributed; 28 invalid questionnaires were eliminated and 344 valid questionnaires were finally obtained.

**Table 1.** Table of evaluation of the effectiveness of PReS policy implementation.

| Evaluating Indicator | Indicator Question Setting | Indicator Options | Definition of Indicator Options | Indicator Properties |
|---|---|---|---|---|
| Family income of resettlers | How much has the total household income increased since the implementation of the PReS policy? | 1 "significantly decreased"; 2 "slightly decreased"; 3 "no change"; 4 "slightly increased or improved"; 5 "significantly increased or improved". | 1: A reduction of more than 20% in total household income;<br>2: Total household income [−5%,−20%);<br>3: Total household income [−5%,+5%);<br>4: Total household income [+5%,+20%);<br>5: An increase of more than 20% in total household income. | Positive Indicator |
| | How much has the per capita disposable income increased since the implementation of the PReS policy? | 1 "significantly decreased"; 2 "slightly decreased"; 3 "no change"; 4 "slightly increased or improved"; 5 "significantly increased or improved". | 1: A decrease of more than 20% in disposable income per capita;<br>2: Disposable income per capita [−5%,−20%);<br>3: Disposable income per capita [−5%,+5%);<br>4: Disposable income per capita [+5%,+20%);<br>5: An increase of more than 20% in disposable income per capita. | Positive Indicator |
| | How much has nonagricultural income increased since the implementation of the PReS policy? | 1 "significantly decreased"; 2 "slightly decreased"; 3 "no change"; 4 "slightly increased or improved"; 5 "significantly increased or improved". | 1: A decrease of more than 20% in the proportion of nonagricultural income;<br>2: The proportion of nonagricultural income [−5%,−20%];<br>3: The proportion of nonagricultural income (−5%,+5%);<br>4: Nonagricultural income ratio [+5%,+20%];<br>5: An increase of over 20% in the proportion of nonagricultural income. | Positive Indicator |
| Living conditions of resettlers | How much have housing conditions improved since the implementation of the PReS policy? | 1 "significantly decreased"; 2 "slightly decreased"; 3 "no change"; 4 "slightly increased or improved"; 5 "significantly increased or improved". | Residential conditions: a: thatched house; b: brick concrete house; c: steel concrete house.<br>1: From c to a;<br>2: From c to b or from b to a;<br>3: No change;<br>4: From a to b or from b to c;<br>5: From a to c. | Positive Indicator |
| | How much has the living environment improved since the implementation of the PReS policy? | 1 "significantly decreased"; 2 "slightly decreased"; 3 "no change"; 4 "slightly increased or improved"; 5 "significantly increased or improved". | Subjective judgment is made by the respondents on the basis of actual situation. | Positive Indicator |
| | How much has the infrastructure improved since the implementation of the PReS policy? | 1 "significantly decreased"; 2 "slightly decreased"; 3 "no change"; 4 "slightly increased or improved"; 5 "significantly increased or improved". | 1: A reduction of more than 20% in the number of infrastructures;<br>2: The number of infrastructures [−5%,−20%);<br>3: The number of infrastructure [−5%,+5%);<br>4: The number of infrastructures [+5%,+20%);<br>5: An increase of more than 20% in the number of infrastructures. | Positive Indicator |
| | How much has the problem been solved since the implementation of the PReS policy? | 1 "significantly decreased"; 2 "slightly decreased"; 3 "no change"; 4 "slightly increased or improved"; 5 "significantly increased or improved". | 1: A Reduction of more than 20% in the number of problems solved;<br>2: Number of problems solved [−5%,−20%);<br>3: Number of problem solved (−5%,+5%);<br>4: Number of problem solved [+5%,+20%);<br>5: An increase of more than 20% in the number of problem solved. | Positive Indicator |

**Table 1.** *Cont.*

| Evaluating Indicator | Indicator Question Setting | Indicator Options | Definition of Indicator Options | Indicator Properties |
|---|---|---|---|---|
| Production conditions of resettlers | How much has the per capita area of arable land improved since the implementation of the PReS policy? | 1 "significantly decreased"; 2 "slightly decreased"; 3 "no change"; 4 "slightly increased or improved"; 5 "significantly increased or improved". | 1: A reduction of more than 20% in arable land per capita;<br>2: Arable land per capita [−5%,−20%);<br>3: Arable land per capita (−5%,+5%);<br>4: Arable land per capita [+5%,+20%);<br>5: An increase of more than 20% in arable land area per capita | Positive Indicator |
| | How much has the per capita grain output increased since the implementation of the PReS policy? | 1 "significantly decreased"; 2 "slightly decreased"; 3 "no change"; 4 "slightly increased or improved"; 5 "significantly increased or improved". | 1: A reduction of more than 20% in grain production per capita;<br>2: Per capita grain production [−5%,−20%);<br>3: Per capita grain production (−5%,+5%);<br>4: Per capita grain production [+5%,+20%);<br>5: An increase of more than 20% in per capita grain production. | Positive Indicator |
| | How much has the level of agricultural mechanization improved since the implementation of the PReS policy? | 1 "significantly decreased"; 2 "slightly decreased"; 3 "no change"; 4 "slightly increased or improved"; 5 "significantly increased or improved". | 1: A reduction of more than 20% in the number of machines and equipment;<br>2: Number of machines and equipment [−5%,−20%);<br>3: Number of machines and equipment [−5%,+5%];<br>4: Number of machines and equipment [+5%,+20%);<br>5: An increase of more than 20% in the number of machines and equipment. | Positive Indicator |
| Local economy | How much has the GDP increased since the implementation of the PReS policy? | 1"significantly decreased"; 2"slightly decreased"; 3"no change"; 4"slightly increased or improved"; 5"significantly increased or improved". | 1: A reduction of more than 20% in GDP;<br>2: GDP [−5%,−20%);<br>3: GDP (−5%,+5%);<br>4: GDP [+5%,+20%);<br>5: An increase of more than 20% in GDP. | Positive Indicator |
| | How much has the number of poor rural resettlers decreased since the implementation of the PReS policy? | 1 "significantly decreased"; 2 "slightly decreased"; 3 "no change"; 4 "slightly increased or improved"; 5 "significantly increased or improved". | 1: A reduction of more than 20% in the number of impoverished people;<br>2: Number of impoverished people [−5%,−20%);<br>3: Number of impoverished people [−5%,+5%);<br>4: Number of impoverished people [+5%,+20%);<br>5: An increase of more than 20% in the number of impoverished people. | Negative indicators |
| | How much has the construction of "one village, one industry" increased or decreased since the implementation of the PReS policy? | 1 "significantly decreased"; 2 "slightly decreased"; 3 "no change"; 4 "slightly increased or improved"; 5 "significantly increased or improved". | 1: A decrease of more than 20% in the number of "one village, one product";<br>2: Number of "one village, one product "[−5%,−20%);<br>3: Number of "one village, one product " [−5%,+5%];<br>4: Number of "one village, one product "[+5%,+20%);<br>5: An increase of more than 20% in the number of "one village, one product". | Positive Indicator |
| | How much has the construction of beautiful homes increased or decreased since the implementation of the PReS policy? | 1 "significantly decreased"; 2 "slightly reduced"; 3 "no change"; 4 "slightly increased or improved"; 5 "significantly increased or improved". | 1: A decrease of more than 20% in the number of Beautiful Home households;<br>2: Number of Beautiful Home households [−5%,−20%);<br>3: Number of Beautiful Home households [−5%,+5%);<br>4: Number of Beautiful Home households [+5%,+20%);<br>5: An increase of more than 20% in the number of Beautiful Home households. | Positive Indicator |

**Table 1.** *Cont.*

| Evaluating Indicator | Indicator Question Setting | Indicator Options | Definition of Indicator Options | Indicator Properties |
|---|---|---|---|---|
| Social stability | How does the number of local letters and visits change since the implementation of the PReS policy? | 1 "significantly decreased"; 2 "slightly decreased"; 3 "no change"; 4 "slightly increased or improved"; 5 "significantly increased or improved". | 1: A reduction of more than 20% in the number of local letters and visits; 2: Number of local letters and visits [−5%,−20%); 3: Number of local letters and visits (−5%,+5%); 4: Number of local letters and visits [+5%,+20%); 5: An increase of more than 20% in the number of local letters and visits. | Negative indicators |
| | How much have complaints changed since the implementation of the PReS policy? | 1 "significantly decreased"; 2 "slightly decreased"; 3 "no change"; 4 "slightly increased or improved"; 5 "significantly increased or improved". | 1: A reduction of more than 20% in the number of complaints; 2: Number of complaints [−5%,−20%); 3: Number of complaints (−5%,+5%); 4: Number of complaints [+5%,+20%); 5: An increase of more than 20% in the number of complaints. | Negative indicators |
| | How does the number of group incidents change since the implementation of the PReS policy? | 1 "significantly decreased"; 2 "slightly decreased"; 3 "no change"; 4 "slightly increased or improved"; 5 "significantly increased or improved". | 1: A reduction of more than 20% in the number of mass incidents; 2: Number of mass incidents [−5%,−20%); 3: Number of mass incidents (−5%,+5%); 4: Number of mass incidents [+5%,+20%); 5: An increase of more than 20% in the number of mass incidents. | Negative indicators |

Note: + means increase; − means decrease.

We adopt stratified sampling in this sampling survey. Samples are taken from high-income resettlement groups, middle-income resettlement groups, and low-income resettlement groups. The income levels in this study are divided on the basis of the per capita income of resettlement households. To be specific, low-income resettlement households with per capita income ranging from 0 yuan to 5303 yuan; middle-income resettlement households with per capita income of 5304 yuan to 15,303 yuan; and high-income resettlement households with per capita income exceeding 15,303 yuan. The income level ratio of sampled resettlement households and the overall income level ratio of resettlement are shown in Figure 4.

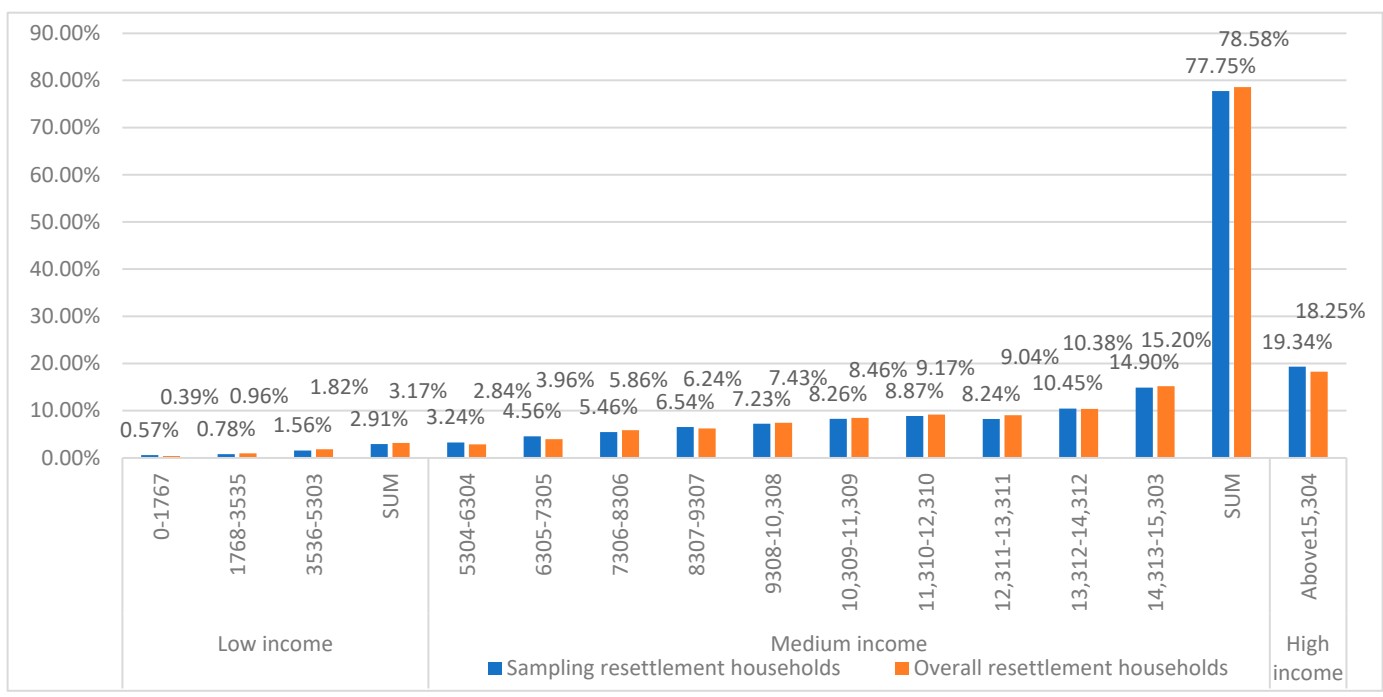

**Figure 4.** Distribution of sampled and overall income levels.

### 2.4. Research Methodology

#### 2.4.1. Structural Equation Model

There are two methods to determine weight: the subjective weighting method and the objective weighting method. The disadvantage of the former is that it is too subjective and depends somewhat on the selected experts. The disadvantage of the latter is that researchers have no discretion and can provide only public factors based on data information. In practical applications, two apparently unrelated factors are often classified into the same category, and researchers cannot correct this mistake. Therefore, this paper selects a method combining subjective and objective factors to determine the index weight by constructing a structural equation. This method can also reflect the structural validity of the questionnaire. We use amos23.0 software to output the model parameter estimation and model goodness of fit test.

#### 2.4.2. Fuzzy Comprehensive Evaluation Method

Because all kinds of qualitative and quantitative impact assessment methods are used to evaluate the implementation effects of the PReS policy mostly from a particular side, the final impact assessment conclusion should be obtained by comprehensive evaluation. There are many kinds of comprehensive impact assessment methods, such as the expert scoring method, AHP, and fuzzy comprehensive evaluation method. However, the basic idea is to comprehensively consider various qualitative and quantitative factors, combine qualitative impact assessment with quantitative impact assessment, and usually use mathematical

processing to quantify qualitative indicators and convert multiple objectives into a single objective to determine the final impact assessment. In this study, the structural equation is constructed to determine the index impact assessment weight, and the implementation effects of the PReS policy are comprehensively evaluated using mathematical fuzzy comprehensive evaluation.

The fuzzy comprehensive evaluation method uses fuzzy membership degree theory to reasonably quantify qualitative indicators and integrates qualitative and quantitative methods. Because many impact assessment factors are involved in the comprehensive impact assessment of the implementation effects of the PReS policy, if only the main factors are considered, some important information will be lost, and the impact assessment results will be distorted. The fuzzy comprehensive evaluation method can better address the problems of multiple factors and qualitative interval impact assessment criteria. To comprehensively consider all of the impact assessment factors, this paper adopts a two-level fuzzy comprehensive impact assessment model. The mathematical fuzzy comprehensive impact assessment model is as follows:

$$B = A \circ R = (b_1, b_2, \ldots, b_n) \tag{1}$$

In the formula, B represents the impact assessment set; $b_n$ represents the membership degree of the survey object to the nth impact assessment; A represents the weight vector; R represents the fuzzy relationship matrix; and $\circ$ represents the fuzzy operation method, which is called the fuzzy operator. The fuzzy operator is an artificial definition, and "$\circ$" can be defined differently to compare which fuzzy operator has the best and most reasonable control effects in the fuzzy reasoning process. There are four common fuzzy operators: "min-max", "product-sum", "min-sum", and "product-max" fuzzy operators. This paper evaluates mainly the implementation effects of the PReS policy. There are many impact assessment index systems, so it is more scientific to choose the product-sum fuzzy operator. This method considers all factors according to weight and is suitable for situations where each factor is effective. For the comprehensive impact assessment of multilevel factors, the multilevel fuzzy comprehensive evaluation method can be used to solve the problem; that is, from the bottom to the top, until the final impact assessment set B is obtained, the k-th impact assessment result is the membership degree of the k-1st factor. Finally, the fuzzy value is removed; that is, the comprehensive impact assessment score W of the impact assessment object is calculated using the fuzzy comprehensive impact assessment set B and the measurement scale H:

$$W = B \times H \tag{2}$$

In Formula (2), H = ("significantly reduced", "slightly reduced", "no change", "slightly increased or improved", "significantly increased or improved") = (1, 2, 3, 4, 5).

The multilevel fuzzy comprehensive impact assessment model is suitable for evaluating the implementation effects of the PReS policy. The steps are as follows. First, five secondary indicators are evaluated. Then, the impact assessment results at this level are taken as the fuzzy relationship matrix of the implementation effects of the PReS policy. Finally, the fuzzy comprehensive impact assessment of the implementation effects of the PReS policy is performed, and the overall comprehensive impact assessment results are obtained. This article defines comments as follows: {[0–1] significantly reduced; [1–2] slightly reduced; [2–3] no change; [3–4] slightly increased or improved; [4–5] significantly increased or improved}.

## 3. Results

### 3.1. Basic Characteristics of the Sample Data

In the questionnaire survey group of the implementation effects of the PReS policy, the survey proportion of county and township cadres, reservoir resettlement cadres, and village cadres is 61.33%, more than half, while the proportion of resettlers surveys is 13.95%, which is relatively small because, during the implementation of the PReS policy, county

and township cadres, reservoir resettlement cadres, and village cadres have done much work, have a good understanding of the PReS policy, and know the problems and effects encountered in implementing the PReS policy, so the proportion of selection is relatively high. In terms of the proportion of educational level, 65.7% are university level or above; this result also corresponds to the high proportion of county and township cadres, reservoir resettlement cadres, and village cadres. Their educational level is generally higher than that of resettlers. For the same reason, from the perspective of current residence, the proportion of villagers' committee residents, township residents, and county residents is 74.41%. There are many ethnic minorities in Zhijin County. In this questionnaire survey, ethnic minorities accounted for 45.35% of the respondents. See Table 2 for details.

**Table 2.** Basic characteristics of the sample data.

| Attribute | Attribute Classification | Sample Value | |
|---|---|---|---|
| | | **Number** | **Proportion (%)** |
| Gender | Male | 221 | 64.24 |
| | Female | 23 | 35.76 |
| Degree of education | Primary school or below education level | 14 | 4.07 |
| | Junior high school education level | 39 | 11.34 |
| | High school/technical secondary school education level | 65 | 18.9 |
| | College education level | 102 | 29.65 |
| | University or above education level | 124 | 36.05 |
| Ethnic groups | Han nationality | 188 | 54.65 |
| | Miao nationality | 19 | 5.52 |
| | Buyi nationality | 5 | 1.45 |
| | Dong nationality | 2 | 0.58 |
| | Yi nationality | 23 | 6.69 |
| | Other ethnic minorities | 107 | 31.1 |
| Current residence | City | 13 | 3.78 |
| | County/town | 75 | 21.8 |
| | Township resident | 121 | 35.17 |
| | Village resident committee | 60 | 17.44 |
| | Villager group | 75 | 21.8 |
| Occupation | Resettlers | 48 | 13.95 |
| | Ordinary farmer | 32 | 9.3 |
| | Village cadres | 86 | 25 |
| | Reservoir resettlement cadres | 10 | 2.9 |
| | County and township cadres | 115 | 33.43 |
| | Other | 53 | 15.42 |

### 3.1.1. Independent Sample Test

Independent samples refer to two groups of samples that are relatively independent of each other. Using an independent sample test, we can determine whether all items can identify the reaction degree of different respondents. In this paper, the samples are first sorted according to the total score of the scale, and the top 30% (high group) and bottom 30% (low group) are obtained. Then, the differences in the high and low groups' answers to each question are tested. According to the output results of SPSS 26.0 software, when the confidence level is 0.05, all indicators pass the independent sample test.

### 3.1.2. Reliability Test of the Questionnaire

The reliability test of the questionnaire refers to the reliability of the questionnaire test results, which reflects mainly the consistency and stability of the questionnaire test results, that is, whether the questionnaire results can reflect the authenticity of the tested samples. Generally, the reliability test essentially tests whether the results of the questionnaire are reliable and whether the tested sample has answered truthfully. Cronbach's alpha is the

most commonly used reliability test method in social science research. In some basic studies, the reliability is acceptable only when it reaches at least 0.80. In some exploratory studies, the reliability is acceptable as long as it reaches 0.70 and the range is between 0.70 and 0.99, indicating high reliability. A reliability lower than 0.5 is low and must be rejected. Using SPSS 26.0 statistical software, we obtain 0.9 as the internal consistency reliability coefficient value of the questionnaire on the "implementation effects of the PReS" policy; this result shows that the internal consistency of the questionnaire is very high, and we calculated the reliability coefficients of the five secondary indicators as 0.727, 0.858, 0.600, 0.742, and 0.684. Therefore, the reliability of all dimensions is acceptable.

*3.2. Weights of Impact Assessment Indicators*

See Figure 5 for the structural equation model for a comprehensive impact assessment of the implementation effects of the PReS policy. See Table 3 for the model-fitting degree test. The overall simulation results of the model are satisfactory. Although the fitting results of the $X^2$/DF and the GFI slightly fail to meet the requirements because of the complexity of social problems, overemphasizing the fitting degree standard is unsuitable for social science research. Steiger proposed that a CFI less than 0.05 indicates a good fit and a CFI less than 0.10 indicates a reasonable fit. A GFI value greater than 0.90 indicates a good fit and a GFI greater than 0.80 indicates that the fit is reasonable. Therefore, the structural equation model for comprehensive impact assessment of the implementation effects of the PReS policy has passed the test. In this paper, the estimated value of the factor load and its significance level are expressed in table form. The significance level passed the test under the condition of 0.01. The load vector can be normalized, and the impact assessment index weight can be obtained after normalization. (Normalization is a dimensionless processing method, which changes the absolute value of coefficient values into a relative value relationship. In this paper, the proportion of the load estimation value of the index factor is taken as its normalized weight.) See Table 4 for specific results.

**Table 3.** Structural equation model fitness test.

| Index | $X^2$/df | RMSEA | RMR | GFI | CFI | NFI | IFI | PGFI | PNFI |
|---|---|---|---|---|---|---|---|---|---|
| Judgment criteria | <3 | <0.08 | <0.05 | >0.9 | >0.9 | >0.9 | >0.9 | >0.5 | >0.5 |
| Value of this structural model | 3 | 0.078 | 0.05 | 0.889 | 0.933 | 0.905 | 0.933 | 0.619 | 0.704 |

**Table 4.** Estimated values and weights of index parameters.

| Primary Index | Secondary Index | Load Estimation of Normalization Factor Numerical Value | Normalized Weight | Third-Level Indicator | Factor Load Estimation Numerical Value | Normalized Weight |
|---|---|---|---|---|---|---|
| Implementation effects of the PReS policy | Family income of resettlers | 0.913 *** | 0.2203 | Total income of resettlers | 0.749 *** | 0.6315 |
| | | | | Per-capita disposable income | 0.218 *** | 0.1838 |
| | | | | Proportion of nonagricultural income | 0.219 *** | 0.1847 |
| | Living conditions of resettlers | 0.769 *** | 0.1856 | Housing conditions | 0.884 *** | 0.2443 |
| | | | | Living environment | 0.901 *** | 0.2490 |
| | | | | Infrastructure | 0.873 *** | 0.2413 |
| | | | | Problem solving | 0.96 *** | 0.2653 |
| | Production conditions of resettlers | 0.763 *** | 0.1841 | Per-capita cultivated land area | 0.141 *** | 0.0827 |
| | | | | Per-capita grain output | 0.831 *** | 0.4877 |
| | | | | Agricultural mechanization level | 0.732 *** | 0.4296 |

**Table 4.** *Cont.*

| Primary Index | Secondary Index | Load Estimation of Normalization Factor Numerical Value | Normalized Weight | Third-Level Indicator | Factor Load Estimation Numerical Value | Normalized Weight |
|---|---|---|---|---|---|---|
| Implementation effects of the PReS policy | Local economy | 0.951 *** | 0.2295 | Gross domestic product | 0.704 *** | 0.2685 |
| | | | | Rural poverty alleviation | 0.535 *** | 0.2040 |
| | | | | One village, one industry | 0.564 *** | 0.2151 |
| | | | | Beautiful home construction | 0.819 *** | 0.3124 |
| | Social stability | 0.718 *** | 0.1733 | Local petition | 0.902 *** | 0.4043 |
| | | | | Appeal change | 0.614 *** | 0.2752 |
| | | | | Mass event | 0.715 *** | 0.3205 |

Note: *** indicates that the parameter is significant at the level of 0.01.

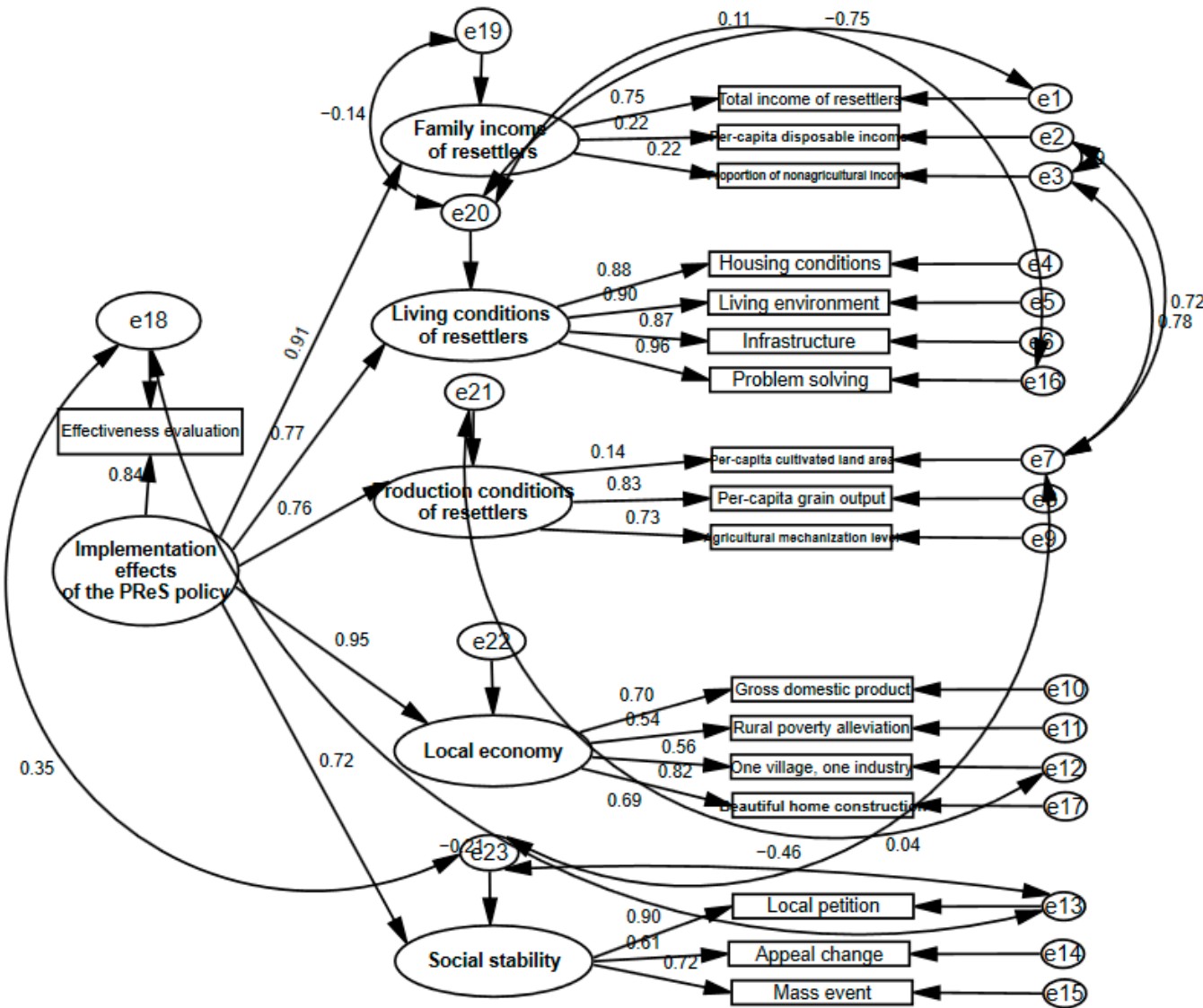

**Figure 5.** Structural equation model for the comprehensive impact assessment of the implementation effects of the PReS policy.

### 3.3. Mathematical Fuzzy Evaluation Results of the Impact Assessment

According to the distribution probability of 344 effective samples' answers to each indicator, the membership degree of the indicator at the corresponding level is determined. For example, for the indicator "total income of resettlers", if 8 of the 344 valid samples answered "significantly decreased", the membership degree of the indicator for the subsystem comments "significantly decreased" is 0.023 (i.e., 8/344 = 0.023). By analogy, the fuzzy relationship matrix of the improvement of five secondary indicators can be constructed. Then, according to the index weight obtained above, the impact assessment result "Wi" of the improvement of each secondary index is calculated: $0 \leq W1 < 1$ is marked as "significantly decreased"; $1 \leq W2 < 2$ is "slightly decreased"; $2 \leq W3 < 3$ is "no change"; $3 \leq W4 < 4$ is "slightly increased or improved"; and $4 \leq W5 \leq 5$ is "significantly increased or improved" (Wi indicates the impact assessment result); the definition of indicator evaluation values is shown in Table 5.

**Table 5.** Definition of index evaluation values.

| Subitem Evaluation Indicators | Definition of Subitem Evaluation Values | Overall Evaluation | Definition of Overall Evaluation Values |
|---|---|---|---|
| Family income of resettlers | W1: A decrease of more than 20% in the income of resettlers families. W2: Income of resettlers families [−5%,−20%]; W3: Income of resettlers families (−5%,+5%); W4: Income of resettlers families [+5%,+20%]; W5: An increase of over 20% in the income of resettlers families. | | |
| Living conditions of resettlers | The living conditions of resettlers are measured from four aspects: living conditions, living environment, infrastructure, and problem-solving. W1: A decrease of more than 20% in the living conditions of resettlers. W2: The living conditions of resettlers [−5%,−20%]; W3: The living conditions of resettlers [−5%,+5%]; W4: Living conditions of resettlers [+5%,+20%]; W5: An increase of more than 20% in the living conditions of resettlers. | | W1: The policy implementation effect is very poor; W2: The policy implementation effect is relatively poor; W3: The policy implementation effect is not significant; W4: The policy implementation has achieved good results. W5: The policy implementation has achieved outstanding results. |
| Production conditions of resettlers | W1: A reduction of more than 20% in the production conditions of resettlers; W2: Production conditions of resettlers [−5%,−20%]; W3: Production conditions of resettlers (−5%,+5%); W4: Production conditions of resettlers [+5%,+20%]; W5: An increase of more than 20% in the production conditions of resettlers. | Overall evaluation of the implementation effect of PReS policy | |
| Local economy | W1: A decrease of more than 20% in local economy; W2: Local economy [−5%,−20%]; W3: Local economy (−5%,+5%); W4: Local economy [+5%,+20%] W5: An increase of more than 20% in local economy. | | |
| Social stability | The level of social stability is measured by the number of local petitions, appeals, and mass incidents. W1: An increase of more than 20% in the number of events; W2: Number of events [+5%,+20%]; W3: Number of events (−5%,+5%); W4: Number of events [−5%,−20%) W5: A decrease of more than 20% in the number of events. | | |

Note: + means increase; − means decrease.

### 3.3.1. Family Income of Resettlers

The impact assessment matrix is as follows:

$$a = A1 \circ R1 = \begin{pmatrix} 0.6315 & 0.1838 & 0.1847 \end{pmatrix} \circ \begin{bmatrix} 0.023 & 0.006 & 0.07 & 0.148 & 0.753 \\ 0.006 & 0.041 & 0.186 & 0.253 & 0.515 \\ 0.02 & 0.081 & 0.224 & 0.183 & 0.491 \end{bmatrix}$$

$$= \begin{pmatrix} 0.019 & 0.026 & 0.120 & 0.174 & 0.641 \end{pmatrix}$$

The membership degrees of the family income of resettlers indicators are 0.019, 0.026, 0.120, 0.174, and 0.641. According to the mathematical fuzzy comprehensive impact assessment Formula (2), the comprehensive impact assessment score is calculated as follows:

$$a = 1 \times 0.019 + 2 \times 0.026 + 3 \times 0.120 + 4 \times 0.174 + 5 \times 0.641 = 4.3,$$
$$4 \le W1 = 4.3 \le 5.$$

The results showed a significant increase or improvement.

### 3.3.2. Living Conditions of Resettlers

The impact assessment matrix is as follows:

$$b = A2 \circ R2 = \begin{pmatrix} 0.2443 & 0.2490 & 0.2413 & 0.2653 \end{pmatrix} \circ \begin{bmatrix} 0.032 & 0.003 & 0.038 & 0.099 & 0.828 \\ 0.023 & 0.006 & 0.044 & 0.119 & 0.808 \\ 0.026 & 0.006 & 0.044 & 0.14 & 0.785 \\ 0.006 & 0.035 & 0.11 & 0.337 & 0.512 \end{bmatrix}$$

$$= \begin{pmatrix} 0.021 & 0.013 & 0.060 & 0.177 & 0.729 \end{pmatrix}$$

The membership degrees of the living conditions of the resettlers indicators are 0.021, 0.013, 0.060, 0.177, and 0.729. According to the mathematical fuzzy comprehensive impact assessment Formula (2), the comprehensive impact assessment score is calculated as follows:

$$b = 1 \times 0.021 + 2 \times 0.013 + 3 \times 0.060 + 4 \times 0.177 + 5 \times 0.729 = 4.6,$$
$$4 \le W2 = 4.6 \le 5$$

The results showed a significant increase or improvement.

### 3.3.3. Production Conditions of Resettlers

The impact assessment matrix is as follows:

$$c = A3 \circ R3 = \begin{pmatrix} 0.0827 & 0.4877 & 0.4296 \end{pmatrix} \circ \begin{bmatrix} 0.02 & 0.061 & 0.259 & 0.241 & 0.419 \\ 0.015 & 0.023 & 0.099 & 0.145 & 0.718 \\ 0 & 0.02 & 0.16 & 0.113 & 0.706 \end{bmatrix}$$

$$= \begin{pmatrix} 0.009 & 0.025 & 0.138 & 0.139 & 0.688 \end{pmatrix}$$

The membership degrees of the production conditions of the resettlers indicators are 0.009, 0.025, 0.138, 0.139, and 0.688. According to the mathematical fuzzy comprehensive impact assessment Formula (2), the comprehensive impact assessment score is calculated as follows:

$$c = 1 \times 0.009 + 2 \times 0.025 + 3 \times 0.138 + 4 \times 0.139 + 5 \times 0.688 = 4.4,$$
$$4 \le W3 = 4.4 \le 5.$$

The results showed a significant increase or improvement.

### 3.3.4. Local Economy

The impact assessment matrix is as follows:

$$d = A4 \circ R4 = \begin{pmatrix} 0.2685 & 0.2040 & 0.2151 & 0.3124 \end{pmatrix} \circ \begin{bmatrix} 0.003 & 0 & 0.093 & 0.218 & 0.686 \\ 0.006 & 0.003 & 0.081 & 0.186 & 0.724 \\ 0.012 & 0.017 & 0.061 & 0.253 & 0.657 \\ 0.003 & 0.032 & 0.055 & 0.256 & 0.654 \end{bmatrix}$$
$$= \begin{pmatrix} 0.014 & 0.014 & 0.072 & 0.231 & 0.678 \end{pmatrix}$$

The membership degrees of the local economy income indicators are 0.014, 0.014, 0.072, 0.231, and 0.678. According to the mathematical fuzzy comprehensive impact assessment Formula (2), the comprehensive impact assessment score is calculated as follows:

$$d = 1 \times 0.014 + 2 \times 0.014 + 3 \times 0.072 + 4 \times 0.231 + 5 \times 0.678 = 4.6,$$
$$4 \leq W4 = 4.6 \leq 5.$$

The results showed a significant increase or improvement.

### 3.3.5. Social Stability

The impact assessment matrix is as follows:

$$e = A5 \circ R5 = \begin{pmatrix} 0.4043 & 0.2752 & 0.3205 \end{pmatrix} \circ \begin{bmatrix} 0.003 & 0.023 & 0.134 & 0.331 & 0.509 \\ 0.012 & 0.003 & 0.262 & 0.212 & 0.512 \\ 0.009 & 0 & 0.247 & 0.151 & 0.593 \end{bmatrix}$$
$$= \begin{pmatrix} 0.007 & 0.010 & 0.205 & 0.241 & 0.537 \end{pmatrix}$$

The membership degrees of the social stability income indicators are 0.007, 0.010, 0.205, 0.241, and 0.537. According to the mathematical fuzzy comprehensive impact assessment Formula (2), the comprehensive impact assessment score is calculated as follows:

$$e = 1 \times 0.007 + 2 \times 0.010 + 3 \times 0.205 + 4 \times 0.241 + 5 \times 0.537 = 4.3,$$
$$4 \leq W5 = 4.3 \leq 5.$$

The results showed a significant increase or improvement.

### 3.3.6. Comprehensive Impact Assessment of the Implementation Effects of the PReS Policy

After calculating the impact assessment matrix of the five secondary indicators, the highest-level comprehensive impact assessment can be performed. The impact assessment matrix is as follows:

$$XG = A \circ R = \begin{pmatrix} 0.2203 & 0.1856 & 0.1841 & 0.2295 & 0.1733 \end{pmatrix} \circ \begin{bmatrix} 0.019 & 0.026 & 0.120 & 0.174 & 0.641 \\ 0.021 & 0.013 & 0.060 & 0.177 & 0.729 \\ 0.009 & 0.025 & 0.138 & 0.139 & 0.688 \\ 0.014 & 0.014 & 0.072 & 0.231 & 0.678 \\ 0.007 & 0.010 & 0.205 & 0.241 & 0.537 \end{bmatrix}$$
$$= \begin{pmatrix} 0.014 & 0.018 & 0.115 & 0.192 & 0.652 \end{pmatrix}$$

The membership degrees of the implementation effects of the PReS indicators are 0.014, 0.018, 0.115, 0.192, and 0.652, respectively. According to the mathematical fuzzy comprehensive impact assessment Formula (2), the comprehensive impact assessment score is calculated as follows:

$$XG = 1 \times 0.014 + 2 \times 0.018 + 3 \times 0.115 + 4 \times 0.192 + 5 \times 0.652 = 4.4,$$
$$4 \leq W = 4.4 \leq 5.$$

The results showed a significant increase or improvement.

In conclusion, the results show that the comprehensive score W1 of the family income of resettlers is 4.3, the comprehensive score W2 of the living conditions of resettlers is 4.6, the comprehensive score W3 of the production conditions of resettlers is 4.4, the comprehensive score W4 of the local economy is 4.6, the comprehensive score W5 of social stability is 4.3, and the comprehensive impact assessment W of the implementation effects of the PReS policy is 4.4, all of which indicate a "significant increase or improvement". See Figure 6 for details.

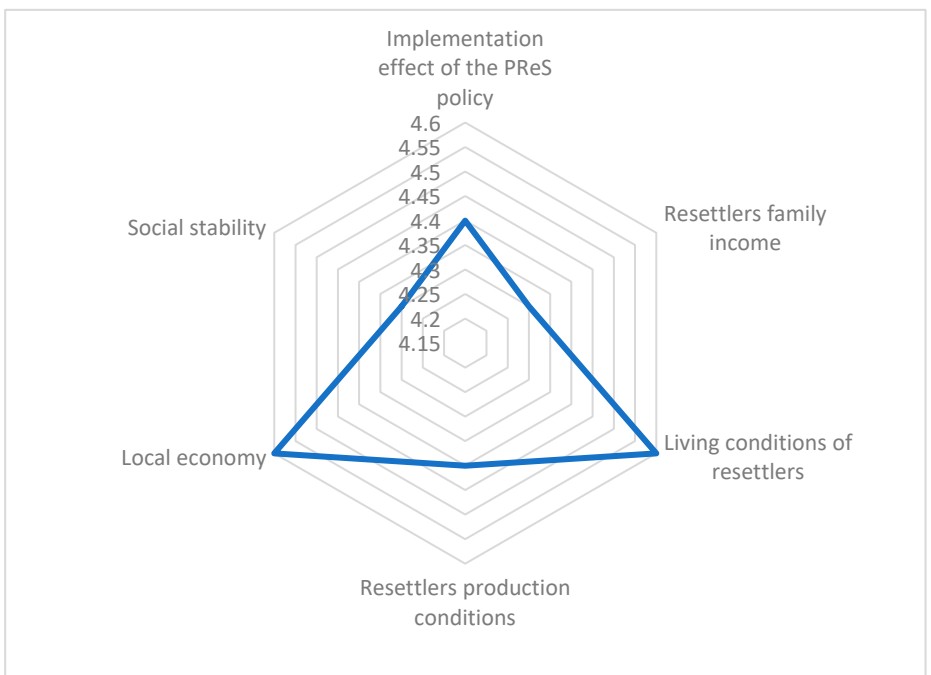

**Figure 6.** Comprehensive impact assessment results of the implementation effects of the PReS policy.

During the author's actual investigation, the most intuitive feeling is that the implementation of the PReS policy has greatly improved the housing conditions of resettlers, as shown in Figures 7 and 8.

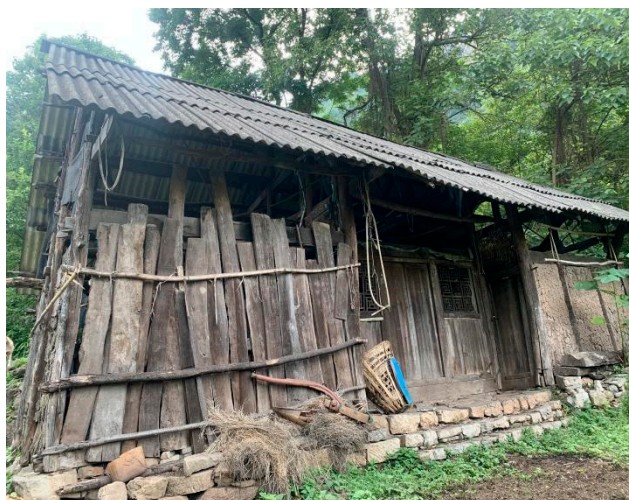

**Figure 7.** House before resettlement (source: on-site shooting by the research team).

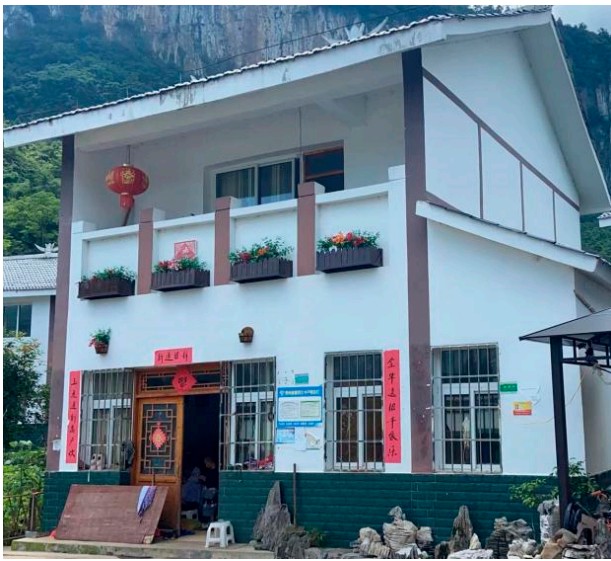

**Figure 8.** House after resettlement (source: on-site shooting by the research team).

## 4. Discussion

### 4.1. Quadrant Analysis of Effectiveness and Importance

In this paper, the effectiveness and importance of the secondary indicators are analyzed using a quadrant chart to identify effective measures to improve the implementation effects of the PReS policy. With the importance of index factors as the abscissa and their effectiveness as the ordinate, the effectiveness values and importance values of all index factors after standardization are positioned on the coordinate axis (Figure 9). In this paper, it is assumed that the area with higher than average effectiveness and importance (weight) is the first quadrant, the area with higher than average effectiveness and lower than average importance is the second quadrant, the area with lower than average effectiveness and importance is the third quadrant, and the area with higher than average importance and lower than average effectiveness is the fourth quadrant. The four quadrants represent the dominant system, the retention system, the observation system, and the improvement system.

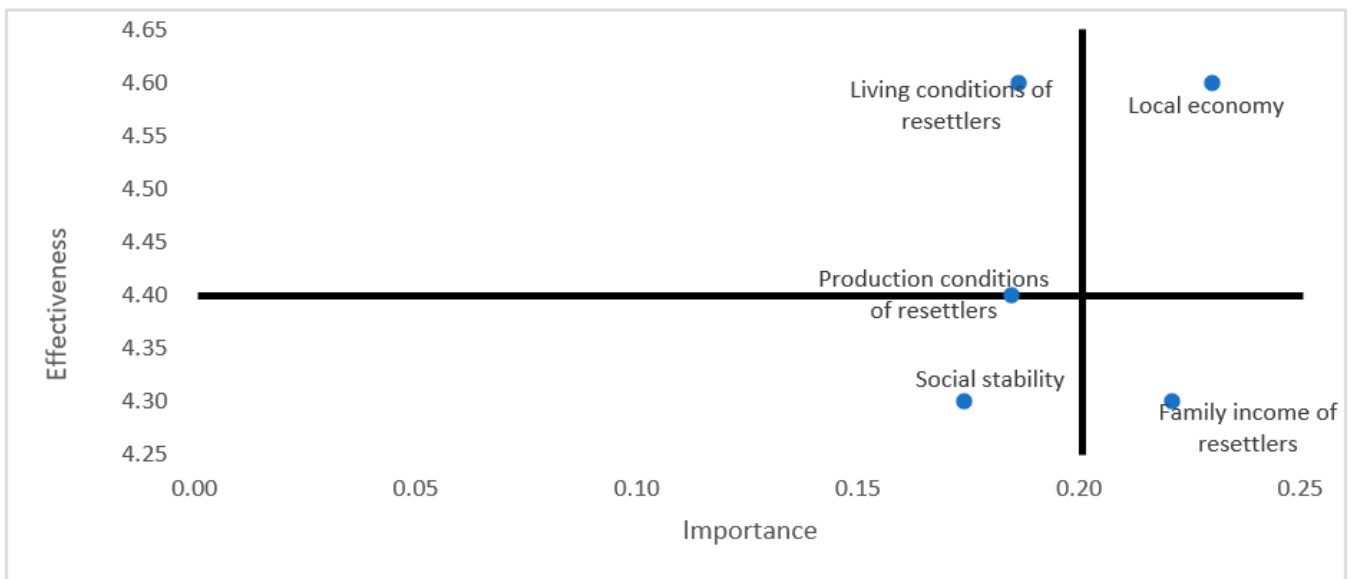

**Figure 9.** Quadrant diagram of importance/effectiveness indicator factors.

### 4.1.1. Dominant System

High effectiveness/high importance. The quadrant chart shows that the importance of local economic development is high and that the implementation of the PReS policy has a strong promotion effects on the development of the local economy.

### 4.1.2. Retention System

High effectiveness/low importance. As the quadrant chart shows, the living conditions and the production conditions of the resettlers significantly improved after the implementation of the PReS policy, which was conducive to the just transition of the resettlers' livelihood and the restoration of the resettlers' livelihood system.

### 4.1.3. Observation System

Low effectiveness/low importance. As the quadrant chart shows, the effectiveness and importance of social stability are relatively low compared with the effectiveness and importance of other secondary indicators; this system needs further observation.

### 4.1.4. Improved System

Low effectiveness/high importance. As the quadrant chart shows, the importance of the resettlers' family income is relatively high, but, after the implementation of the PReS policy, the resettlers' family income is not as significant as other factors; this finding indicates that more efforts should be made to increase the immigrants' income.

### *4.2. Exploring the Experience of Effective Implementation of the PReS Policy*

According to the above comprehensive impact assessment results, the implementation of the PReS policy in Zhijin County, Guizhou Province, China, is very effective, and the resettlers' income, living conditions, production conditions, local economy, and social stability have been significantly improved, effectively realizing a just transition. The results show that the PReS policy has played an important role in restoring the livelihood of resettlers, increasing their income, improving their lives, and sharing development benefits [32,33]. The PReS policy has been effective because, since the implementation of the policy in 2006, Zhijin County has not only completed the phased objectives of support for reservoir resettlement as scheduled, but also better embedded the production support plan for reservoir resettlement into the county's rural revitalization strategy and local development planning [34], which has the following three main components.

### 4.2.1. Poverty Alleviation Strategy

Zhijin County takes poor resettlers' villages as the main support objects; accelerates infrastructure construction, industrial development, poverty alleviation training, and so on; and promotes the development of industrial poverty alleviation and social poverty alleviation. In implementing the reservoir resettlement industry poverty alleviation project, aquaculture practitioners have been trained and cooperatives have been developed to lift all poor resettlers in the reservoir area out of poverty and promote the development and upgrading of the reservoir resettlement industry. In addition, Zhijin County attaches special importance to education support and implements the resettler quality improvement project, which reduces the dropout rate and the economic burden of family education expenditures [35,36].

### 4.2.2. Beautiful Home Construction Strategy

Zhijin County makes use of administrative villages with certain industrial foundations; good ecological environments; convenient transportation; and characteristic villages with good natural landscape resources, traditional cultural resources, and industrial characteristic resources to carry out the construction of beautiful and livable demonstration villages. Additionally, by strengthening infrastructure construction, the infrastructure level of the reservoir area and the resettlement area has been further improved, laying a good

foundation for rural revitalization and enabling the reservoir resettlement policy to be embedded into the local rural revitalization plan [37,38].

### 4.2.3. Rural Revitalization Construction Strategy

Zhijin County pays attention to the upgrading of the reservoir resettlement industry; takes the implementation of the rural revitalization strategy as an opportunity; makes good use of the business cards promoting the county as the "hometown of bamboo fungus in China" and "the largest distribution center of Gleditsia sinensis in China"; highlights "demonstration sites"; and plans county and township demonstration sites in accordance with the principle of "resource integration, capital concentration, and technological innovation". In promoting rural revitalization, Zhijin County adheres to adapting measures to local conditions by taking precise measures according to the actual situation of townships (subdistricts) and villages; actively adjusting the industrial structure through the guidance of township (subdistrict) party committees and grassroots party branches; vigorously guiding villages to take the development path of "one village, one industry"; powerfully driving resettlers to increase income and become rich; and boosting rural revitalization [39,40].

## 5. Conclusions

In this paper, a comprehensive impact assessment index system for the implementation effects of the PReS policy is constructed. The structural equation model is used to determine the weight of the impact assessment index and the fuzzy comprehensive evaluation method is used to comprehensively evaluate the implementation effects of the PReS policy. The impact assessment results show that the comprehensive impact assessment score of the implementation effects of the PReS policy is 4.4, and the effectiveness is very significant. The comprehensive score of the family income of resettlers is 4.3, the comprehensive score of the living conditions of resettlers is 4.6, the comprehensive score of the production conditions of resettlers is 4.4, the comprehensive score of the local economy is 4.6, and the comprehensive score of social stability is 4.3, all of which indicate a "significant increase or improvement". This finding shows that the PReS policy has played an important role in the just transition and restoration of resettlers' livelihood, income improvement, life improvement, and sharing of development benefits. China's PReS policy has far-reaching significance. First, the policy helps resettlers improve their production and living conditions. The direct purpose of the PReS policy is to solve the lack of food and clothing for the resettlers and the weak infrastructure construction in the reservoir area and the resettlement area, improve the production and life of the resettlers, and promote economic development in accordance with the policy of development resettlement. Second, the PReS policy is the inherent requirement of the comprehensive poverty alleviation and rural revitalization strategy. We will improve the implementation of the PReS policy; ensure that resettlers can move in, stay stable, and prosper; consolidate and enhance the results of poverty alleviation; and win the battle against poverty. Third, the policy is conducive to building a socialist harmonious society. Fourth, the policy is conducive to the further development of water conservancy and hydropower [41–43]. Therefore, we provide some suggestions for the government that can further improve the reservoir resettlers' living quality: (1) improving the capacity level of government resettlement agencies; (2) strengthening dynamic management of the resettlement population; (3) strengthening the implementation of project management; (4) exploring differentiated support strategies; (5) increasing investment in advantageous industries and providing more job opportunities for resettlers; and (6) strengthening fund management effectively.

**Author Contributions:** B.L.: Full text writing: G.S.: Paper framework construction: Y.W.: Manuscript revision: Z.S.: Method guidance; B.Z.: Software drawing; Y.X.: Provide data and information; Y.D.: Participate in research. All authors have read and agreed to the published version of the manuscript.

**Funding:** This work was Supported by "Postgraduate Research & Practice Innovation Program of Jiangsu Province: Climate Migration Types and Risk Management in Coastal Areas (grant number: 422003151)"; The Fundamental Research Funds for the Central Universities: Climate Migration Types and Risk Management in Coastal Areas (grant number B230205032); and The Key Research Project of the National Foundation of Social Science of China: Community Governance and Post-relocation Support in Cross District Resettlement [grant number 21&ZD183].

**Data Availability Statement:** The data presented in this study are available on request from the corresponding author. The data are not publicly available due to restrictions e.g., privacy.

**Acknowledgments:** Thank you for the scientific research platform provided by Hohai University, the valuable revision opinions put forward by the review experts, and the people who provided help in the process of writing the paper.

**Conflicts of Interest:** The authors reported no potential conflict of interest.

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
