# Peer review of "Impact Assessment of the Implementation Effect of the Post-Relocation Support Policies of Rural Reservoir Resettlers’ Livelihoods in Energy Transition"

_water, doi:10.3390/w15173129_

Round 1
Reviewer 1 Report
Thanks the authors for focusing on the rural reservoir resetlers. The data and analysis are reliable, but some concerns are still necessary to discuss, as follows:
1. “ the energy system must be changed globally to replace fossil fuels.” Is there any proof for this sentence? Otherwise, I think it is not so prudent.
2. The poverty incidence rate is from 2014, nearly ten years ago. It is better to provide latest data.
3. Where were the resetlers from and When did they come to Zhijin?
4. The time since they moved to Zhijin may also impact there living quality. How do the authors consider this impact?
5. Is there anything can be improved to further promote the reservoir resetlers living quality? Can the authors provide some suggestions for the government?
Author Response
We are very grateful for your suggestions for revisions to the manuscript, and we elaborate on the specific changes below (See manuscript revision model for details).
1.“ the energy system must be changed globally to replace fossil fuels.” Is there any proof for this sentence? Otherwise, I think it is not so prudent..
Response: Thank you for pointing out the problem, we have removed the sentence.
2.The poverty incidence rate is from 2014, nearly ten years ago. It is better to provide latest data.
Response: Zhijing County started the poverty building card in 2014. 2020, they have all been out of poverty, the poverty rate is 0. So the poverty rate here refers to the poverty rate in 2014 when the poverty building card was established, which is just a background.To avoid ambiguity, the relevant description has been removed.
3.Where were the resettlers from and When did they come to Zhijin?
Response: The resettlers are local indigenous residents of zhijin who had to be resettled back in the prefecture due to the construction of the reservoir, which flooded their original houses. the PReS policy was implemented in 2006, so the start year of the policy assessment is 2006.
4.The time since they moved to Zhijin may also impact there living quality. How do the authors consider this impact?
Response: These resettlers are indigenous residents who are resettled in a backward rehousing manner, according to the original size and standard policy for resettlement. The PReS policy was implemented in 2006 after their relocation, and the policy was evaluated from 2006 to 2022.
5.Is there anything can be improved to further promote the reservoir resetlers living quality? Can the authors provide some suggestions for the government?
Response: so, we provide some suggestions for the government which can be improved to further promote the reservoir resettlers living quality :(1)Strengthening the capacity level of government resettlement agencies; (2) Strengthening dynamic management of resettlement population;(3) Strengthening the implementation of project management;(4) Exploring differentiated support strategies; (5) Increasing investment in advantageous industries and provide more job opportunities for resettlers; (6) Strengthening fund management effectively.

Author Response
We are very grateful for your suggestions for revisions to the manuscript, and we elaborate on the specific changes below (See manuscript revision model for details).
1.Line 129-132: the meaning of “one try” still need to be clarified;
Response: one try” have been clarified: "one try" means that the government should try to provide resettlers directly cash compensation as much as possible;
2.Line 134: “solve” is not a suitable word, which need to be replaced by “study” or “assess”, because no paper can solve this problem of resettlers;
Response: Thank you for your modification suggestions. We have replaced “solve” with“study”;
3.Line 203-212: there are 17154 (assuming all immigrants are settlers) resettlers in Zhijin County? Is the number of 344 samples enough to evaluate the issue? From statistical point of view, how many samples are enough to describe or represent the resettlers issue in Zhijin County? Further in Guizhou?
Response: From a statistical perspective, there is a formula for calculating the sampling ratio. Generally, a sampling ratio of 1% to 10% is more appropriate. Of course, the higher the sampling ratio, the better. Our sampling ratio here is 2%. During the survey, due to the impact of the COVID-19, our sampling ratio was relatively low. In order to make the sample more representative, on the one hand, we adopted Stratified sampling, and selected resettlers with relatively high, medium and low incomes for the questionnaire survey. On the other hand, we conducted in-depth interviews with resettlers.
4.Line 247: How weight A was determined?
Response: Firstly, the Likert five scale method was used for our questionnaire. Secondly, the structural equations were constructed in AMOS software, and then the questionnaire data were imported into AMOS software and calculated to obtain the structural equation diagram, see Figure 4. finally, the weights were obtained by normalizing the road strength coefficients in the structural equation diagram, see Table 4. the calculation results of weight A are shown in Section 3.2.
5.Line 273 - 275: as for “{[0-1) significantly reduced; [1- 2 )slightly reduced; [2-3)no change; [3-4) slightly increased or improved;[4-5] significantly increased or improved}”, can you give a relative number to define how much is significant or insignificant? For example, income increase 100% from 1000 Yuan to 2000 Yuan per year compare to previous income is significant.
Response: In order to construct structural equations to find the evaluation index weights, the questionnaire was asked to use the Likert five scale method, described in detail in 2.3.1. the final evaluation values ranged from 1 to 5, described in detail in 3.3. the evaluation values from 1 to 5 here are evaluation values, not specific values.
6. In Table 2, the occupation of the resettlers: the proportion of cadres is 211/344=61% of the whole samples. Do you think it is reasonable and be representative for whole resettlers? Or the resettlers are resettled as local leaders? The final results will be biased to Cadres because 61% of whole resettlers studied are from cadres.
Response: Our article explores the effects of PReS policy implementation. Cadres is the specific operator of PReS policy implementation, and they also evaluate the effects of PReS policy implementation every year and have a lot of data.reservoir resettlement cadres and village cadres have done much work, have a good understanding of the PReS policy, and know the problems and effects encountered in implementing the PReS policy, so the proportion of selection is relatively high. We have also conducted a large number of interviews with resettlers as a supplement to the evaluation of the effectiveness of PReS policy implementation.
7.In the whole of resettlers, please give the proportion of different occupations in Zhijin.
Response: Hydroelectric power plants are generally built in mountainous areas with abundant water resources, and this paper studies the effect of PReS policy implementation for rural reservoir settlers. Here the rural reservoir settlers refer to the inundated mountain farmers, who all make their living from agriculture. the PReS policy does not include urban residents.
8. Explanation on the level of English writing in this article
We have found a professional third-party language polishing company AJE to polish the English of this article. And invited a native English teacher to check the language of this article.

Round 2
Reviewer 2 Report
Sampling is a problem, but I strongly suggest you to add a section of Limitation of the Study to explore the effects of biased sampling on the final results.

The English is OK in a whole, but for something like "One Try", "Two Can", the explanation/definition is in terrible English, which needs to be improved.
Author Response
Thank you for your valuable modification suggestions. We have had a serious discussion. We have responded and made modifications to your proposed modification suggestions.
- We have submitted the correct word and PDF versions of the manuscript;
- Regarding this issue, according to your revision suggestions, we have added a limitation section in our manuscript to explain it;
- According to your modification suggestions, we have drawn a histogram in section 2.3.2;
- According to your modification suggestions, we have added Table 5 in Section 3.3 to define the fuzzy evaluation values;
- According to your modification suggestions, we have defined the values between 1-5 in Table 1 of Chapter 2.3;
- According to your modification suggestions, we have added research limitations before the conclusion;
- Table 1 is not actually a questionnaire, but an evaluation table for the effectiveness of PReS. We have made modifications to the relevant expressions.

Round 3
Reviewer 2 Report
1. The figure 4 is not acceptable. the distribution has to be with 10 to 15 bins to show, not with 3 groups.
2. there must be actual analysis of limitations in this research, not only several sentence. If you cannot do such an analysis of limitations due to your sampling bias, I will reject it next time.
English is OK.
Author Response
We express our sincere gratitude to your valuable revisions, which have made our manuscript more comprehensive.
1.We have redrawn Figure 4 according to your suggestion.
2.We conduct sample sampling based on the number of households. The overall population is 5077 households, with a sample of 344 households and a sampling ratio of 6.8%. According to your modification requirements, we have provided three specific explanations on the limitations of our research.
We would like to express our gratitude for your meticulous suggestions and hope that with your guidance and assistance, we can publish the research together. We would greatly appreciate it.

Round 4
Reviewer 2 Report
I have made your data and method show your results are highly influenced by your sampling bias. Of course, your conclusions are thus biased and not reliable. As I mentioned, if you can provide actual analysis of the effects of sampling bias on your results, or sensitivity analysis of of your model with respect to the resettlers' income, there will be an opportunity to accept this paper. However, with the current state of this manuscript, it must be rejected.
The biased sampling definitely affects the conclusions of this manuscript. The authors hid their sampling bias. although I identified such a sampling bias, an opportunity was still open to them. However the authors didn't make substantial improvement with enough analysis of effects of sampling bias. I have to reject this paper.